

# Influence of Saharan dust on Atlantic tropical cyclones
**Zhenxi Zhang[1] and Wen Zhou[2]**
[1] College of Energy and Power Engineering, Inner Mongolia University of Technology, Hohhot
010010, China. Email: xizhenz690400.student@sina.com
[2] School of Energy and Environment, City University of Hong Kong, Hong Kong, Tat Chee
Avenue, Kowloon, Hong Kong 999077, China. Email: wenzhou@cityu.edu.hk
*Correspondence to*: Zhenxi ZHANG (xizhenz690400.student@sina.com)
## Abstract
The influence of Saharan dust outbreaks on summertime Atlantic tropical cyclone (TC)
activity is explored using continuous atmospheric reanalysis products and TC track
data from 1980 to 2019. Analyses reveal that the Saharan dust plume over the tropical
Atlantic can affect TC activity by affecting the atmospheric hydrology and radiation
absorbed by the earth's surface, which can be classified into three mechanisms. (1) A
strong Saharan dust plume indirectly induces the reduction of atmospheric moisture,
which further suppresses TC track, number of TC days, and intensity, with the
influence covering the whole tropical Atlantic. (2) A strong Saharan dust plume
enhances atmospheric moisture just along the North Atlantic ITCZ through the dust
microphysical effect, which further promotes TC activity along 10ºN latitude in June.
(3) The climatological influence of dust on TC activity is caused by the strong
radiative forcing of Saharan dust over the eastern tropical Atlantic in June, which
produces an evident reduction in SST and lessens the duration and intensity of
regional TC activity in June, according to the 40-yr average from 1980 to 2019.



## 1. Introduction

Atlantic tropical cyclones (TCs) are becoming more destructive economically, as evidenced by the fact that five of the ten most expensive storms in United States history have occurred since 1990. According to the World Meteorological Organization, the societal impact TCs has recently increased for the rising populations and infrastructure in coastal regions.

When easterly trade winds pass over the Saharan Desert, dust and dry air mix to form a layer called the Saharan air layer (SAL), which extends westward from West Africa to the North Atlantic, with easterly trade winds in the tropics, and occurs over extensive portions of the northern tropical Atlantic Ocean. Observational analysis and numerical simulation have suggested that the formation and intensity of TCs in the Atlantic are influenced by the SAL (Dunion and Velden, 2004; Wu et al., 2006; Evan et al., 2006). The SAL prevents TCs from intensifying into mature hurricanes because they are highly negatively correlated with each other. Wu (2007) further investigated the role played by the SAL in long-term changes in TC intensity. Dunion and Velden (2004) attributed the interaction between TCs and the SAL to fluid dynamical mechanisms, e.g., vertical wind shear and suppression of deep convection caused by the SAL. However, less attention has been paid to the relationship between TC activity and the impact of dust on the atmospheric hydrology and radiation absorbed by the earth's surface.

The SAL transports large plumes of Saharan dust across the northern tropical Atlantic. As absorbing aerosols, mineral dust absorbs solar radiation to heat the atmosphere and enhance cloud evaporation, known as the semi-direct effect (Huang et al., 2006; Lau et al., 2009). In microphysics, mineral dust particles are effective cloud condensation nuclei (Koehler et al., 2009; Karydis et al., 2011) and ice nuclei (Chen et al., 1998; Hoose and Mohler, 2012; Cziczo et al., 2013). Saharan dust, especially, has been found to enhance ice cloud during its transport across the Atlantic (DeMott et al., 2003; Sassen, 2003; Cziczo et al., 2004). Aerosols can absorb and scatter solar





radiation, leading to a large reduction in the solar radiation absorbed by the earth's
surface, which is referred to as direct radiative forcing (Ramanathan et al., 2001).
Through direct radiative forcing, dust particles can cool the surface (Cavazos et al.,
2009). Numerical experiments on a Saharan dust storm suggest that strong radiative
forcing of dust particles can reduce surface temperature by 0.2-0.5 K over most of
western Europe (Bangert et al., 2012).
The objective of this study is to investigate the relationship between Atlantic TC
activity and the changes in atmospheric hydrology and surface temperature caused by
Saharan dust. Data are described in section 2. The impacts of Saharan dust on the
atmospheric hydrology and sea surface temperature (SST) over the tropical Atlantic
are discussed in sections 3 and 4, respectively. The influence of Saharan dust on
Atlantic TC activity is analyzed in section 5. A summary and conclusions are
presented in section 6.
**2. Datasets**
The Modern-Era Retrospective Analysis for Research and Applications (MERRA) is a
satellite-era atmospheric reanalysis (Rienecker et al., 2008) that focuses on analyses
of the global hydrological cycle regarding precipitation and water vapor climatology
(Rienecker et al., 2011). The new version of MERRA, MERRA-2, improves the
computation of the hydrological cycle (Takacs et al., 2015; Reichle and Liu, 2014) by
not only incorporating new observations but also reducing spurious trends and jumps
caused by changes in meteorological observations (McCarty et al., 2016).
It is significant that MERRA-2 includes analyzed aerosol fields for the first time, to
allow the investigation of aerosol-climate or aerosol-weather interactions (Bellouin et
al., 2013; Reale et al., 2014). Aerosol simulation in MERRA-2 is implemented with a
radiatively coupled version of the Goddard Chemistry, Aerosol, Radiation, and
Transport model (GOCART, Chin et al., 2002; Colarco et al., 2010). The Aerosol
Optical Depth (AOD) and other observable aerosol properties simulated with the
GOCART aerosol module have been validated by numerous studies (e.g., Colarco et



al., 2010; Nowottnick et al., 2010, 2011; Bian et al., 2013). On the other hand, aerosol
fields in MERRA-2 assimilate the satellite-observed AOD using the Advanced Very
High Resolution Radiometer (AVHRR) (Heidinger et al., 2014) and the Multi-angle
Imaging SpectroRadiometer (MISR) (Kahn et al., 2005), as well as ground-based
measurements of AOD from the AErosol Robotic NETwork (AERONET) (Holben et
al., 1998). The MERRA-2 data used in this study are the monthly dust AOD, liquid
water path (LWP), ice water path (IWP), and sea surface temperature (SST) for
1980-2019, with a resolution of 0.625º longitude by 0.5º latitude. LWP and IWP are
the vertical integration of the liquid and ice water in the air column, respectively.
Atlantic TC track data for 1980-2018 are obtained from NOAA's Tropical Prediction
Center. TC data in 2019 are obtained from the National Hurricane Center (NHC)
Hurricane Best Track Files. All the data are recorded at six-hour intervals, and
missing data are indicated by zeros. The parameters contained in the data include the
month, day of the month, hour (GMT), latitude (degrees), longitude (0-360 degrees),
maximum wind speed (m s$^{-1}$), and central surface pressure (hPa).
**3. Association patterns of LWP and IWP with dust**
In this section, we discuss the impact of Saharan dust on LWP and IWP over the
Atlantic through the semi-direct and microphysical effects. The observed dust AOD
over the North Atlantic peaks in the summer (Kaufman et al., 2005), while the
Atlantic hurricane season occurs mainly in summer. Figure 1 is the 40-yr averaged
(1980-2019) monthly dust extinction AOD from June to September, which presents
the monthly variation in the transport of large plumes of Saharan dust across the
northern tropical Atlantic.
**3.1 LWP**
Figure 2 shows the 40-yr (1980-2019) average of LWP during the summer.
Climatologically, LWP occurs over the Atlantic ITCZ and the western and central
tropical Atlantic, with the maximum appearing along the ITCZ between 20ºW and



50ºW, which shows a descending tendency in intensity and area from June to
September. An absence of LWP occurs over the eastern tropical Atlantic, collocated
with regions of heavy dust loading, and the region of LWP absence migrates eastward
and northward from June to September, corresponding to the monthly variation in the
westward extension of the Saharan dust plume.
To study the long-term statistical relationship between dust AOD and LWP,
inter-annual correlation is computed between the monthly averaged dust AOD and
LWP at each grid point during 1980-2019, which is also shown in Figure 2. This
analysis reveals that LWP has significant negative correlations with dust AOD in two
parts of the tropical Atlantic. One is in the Caribbean Sea and appears in June.
Another is in the eastern tropical Atlantic where the gradient of LWP is very large, as
can be seen in August and September. Marine areas with a significant negative
correlation ($< -0.5$) are larger in August than in other months.
Although correlation computations present the negative response of LWP to dust
AOD, this is a qualitative result. And this response does not indicate a complete linear
relationship according to the value of the correlation coefficient. On the other hand,
the impact of strong dust storms is significantly underestimated by the climatology of
the dust plume (Figure 1), which represents just the background levels. Therefore, 8
years with the strongest and weakest dust AOD (averaged over the region of
10ºN-25ºN, 60ºW-20ºW) are selected during 1980-2019 to construct the strongest and
weakest dust conditions, respectively. Table 1 gives the magnitude of change in the
strongest and weakest dust AOD compared to climatology.
The magnitude of the changes in LWP in the 8-yr strongest and weakest dust
conditions is shown in Figure 3. The LWP difference map of strong-minus-mean
shows that the strengthening Saharan dust plume continuously suppresses LWP from
June to September, with the most pronounced suppression of 0.025-0.075 kg m$^{-2}$
appearing at several locations of the dust plume zone over the eastern tropical Atlantic.
These locations are basically the same as the response regions shown by the grid point
correlation computation (Figure 2). The regions with reduced LWP migrate eastward



from the Caribbean Sea in June to offshore of the continent in September,
corresponding to the monthly variation in the westward extension of the dust plume.
September has the largest response region of LWP of any month to the strengthening
dust plume, with a zonal range from 5ºN to 35ºN. On the other hand, LWP is
enhanced when the Saharan dust plume becomes weak, as shown by the LWP
difference of weak-minus-mean. The magnitude of the increase is greatest in July, in
the range of 0.05-0.15 kg m$^{-2}$. The response of LWP in September is weak because the
region with enhanced LWP is very scarce and small. The negative response of LWP to
dust AOD is indicated not only qualitatively by correlation computations, but also
quantitatively in the extreme dust conditions. Apparently, this feature is robust
because it is independent of the analysis method and data sample. The analysis in
Figures 2 and 3 reveals the details of the semi-direct effect of Saharan dust on LWP
over the Atlantic, including temporal variation, spatial distribution, and magnitude of
change.
The warm and dry air masses in the SAL originate from the west coast of Africa and
extend westward to the tropical Atlantic. The mixture of these warm and dry air
masses with cool and wet marine air masses can reduce the humidity of the
atmosphere, inducing an absence of LWP over the eastern tropical Atlantic (Figure 2).
However, the reduction of LWP shown in Figure 3 occurs with strong dust conditions
and is located over the western tropical Atlantic, coinciding with the top of the dust
plume tongue. This demonstrates that the reduction of LWP shown in Figure 3 is
associated with the dust semi-direct effect, not the warm and dry air masses in the
SAL.
**3.2 IWP**
Figure 4 shows the climatology pattern of IWP. Compared to the LWP pattern (Figure
2), IWP distributes over the Atlantic ITCZ and the western tropical Atlantic. MODIS
observations indicate that there is no cirrocumulus over the eastern tropical Atlantic,
where there is a dust storm (Figure 3 in Kaufman et al., 2005). As with the monthly





variation of LWP, the region of absence of IWP over the eastern tropical Atlantic
migrates eastward and northward from June to September. Figure 4 also displays the
correlation between dust AOD and IWP at each grid point. IWP in the southern
Caribbean has significant negative correlations with dust AOD in June, the same as
the LWP pattern (Figure 2). IWP over the African Sahel and adjacent offshore region
consistently shows negative correlations with dust AOD from June to September, and
this association of IWP with dust AOD becomes strongest in August because of the
strongest negative correlation coefficient (< -0.7) and largest impact area.
Figure 5 shows the difference in IWP between the extreme dust conditions and 40-yr
average, which is analyzed according to the dust semi-direct effect firstly. In June, the
IWP difference between the 8-yr strongest dust conditions and the 40-yr average
shows that the intensification of the dust plume is accompanied by a reduction of IWP
in the region of the dust plume tongue over the western Atlantic, with the magnitude
of change in IWP being 0.05-0.15 kg m$^{-2}$. A weakened dust plume cannot affect the
same region where IWP is almost the same as the 40-yr average, without obvious
change, as shown by the IWP difference between the 8-yr weakest dust conditions and
the 40-yr average. In the strongest dust conditions in August and September, there is a
reduction of 0.05-0.15 kg m$^{-2}$ in IWP over the African Sahel and adjacent offshore
region compared to the 40-yr average. In contrast, IWP increases by 0.05-0.1 kg m$^{-2}$
in the weakest dust conditions, but this enhancement appears only in very small areas
offshore of the African Sahel. Besides showing the magnitude of the change in IWP,
Figure 5 indicates that the features presented above are consistent with the results of
the linear correlation analysis (Figure 4).
Besides the semi-direct effect, the difference in IWP shown in Figure 5 is also
analyzed according to the dust microphysical effect. In June and July, directly south of
the dust plume, IWP along the North Atlantic ITCZ is enhanced by 0.05-0.3 kg m$^{-2}$
(0.05-0.25 kg m$^{-2}$ in June; 0.05-0.3 kg m$^{-2}$ in July) in the strongest dust condition and
reduced by 0.05-0.15 kg m$^{-2}$ in the weakest dust condition. The positive relationship
of IWP with dust in July even appears in northern South America. This response of



IWP to dust is similar to observations (Wilcox et al., 2010) and simulations (Lau et al.,
2009) of enhanced summer precipitation along the ITCZ during dust outbreaks.
A comparison between Figures 3 and 5 shows that the response area of LWP to dust is
evidently larger than that of IWP, indicating a stronger dust semi-direct effect on LWP
than on IWP. Attributed to dust radiative heating (Carlson and Benjamin, 1980; Alpert
et al., 1998; Zhu et al., 2007; Wong et al., 2009), the transport of the Saharan dust
plume is accompanied by significant warming between 900 and 600 hPa (Wilcox et
al., 2010), and its impact on liquid water cloud in the lower troposphere is larger than
on ice cloud in the upper troposphere.

### 204    4. Association patterns of SST with dust

Figure 6 shows the climatology of SST over the tropical Atlantic. Besides the
northward decrease from the ITCZ, the most pronounced feature of SST distribution
is the eastward decrease, with the maximum in the Gulf of Mexico and Caribbean Sea,
and the minimum offshore of Africa. This zonal variation of SST indicates an
eastward migration tendency from June to September. A grid point correlation
between SST and dust AOD is also presented in Figure 6. Because only the strong
radiative forcing of dust is associated with a reduction in surface temperature (Bangert
et al., 2012), the correlation coefficients are computed over the dust plume regions
where dust AOD is larger than 0.15. SST consistently shows negative correlations
with dust AOD from June to September, with the largest response area in June.
Although the association of SST with dust AOD is the same as for LWP and IWP, the
degree of the negative correlation (coefficient can be -0.5) is smaller than for LWP
and IWP (coefficient can be -0.7).
Figure 7 shows the difference in SST between the extreme dust conditions and the
40-yr average. It is in June and September that SST presents an evident response to
dust AOD. Over the dust plume regions, there is a general reduction in SST under the
strong dust condition compared to the 40-yr average, and an increase in SST under the
weak dust condition (the increase in June is not shown, because its magnitude is





smaller than 0.2 °C). The most pronounced response of SST to dust appears in June,
because the reduction in SST in June (0.2-0.6 °C) is twice the magnitude of that in
September (0.1-0.3 °C). The analyses in Figures 6 and 7 reveal the impact of dust on
SST through direct radiative forcing, including temporal variation and spatial
distribution. Table 2 lists the magnitude of change in SST averaged over the region of
10ºN-25ºN, 60ºW-20ºW, corresponding to the strong and weak dust conditions,
respectively. In June, the reduction of SST related to strong dust radiative forcing
accounts for 1.22% of the 40-yr average, which is larger than that in any other month.

## 5. Association patterns of TCs with dust

Atlantic TC statistics are obtained by summing the total number of TC days (and
intensity) in a 4-degree grid cell. Figures 8 and 9 show the regions with TC days and
intensity statistics in the strong and weak dust conditions. A comparison of TC
statistics between the strong and weak dust conditions reveals that the strengthening
Saharan dust plume can suppress TC duration and intensity. In detail, this suppression
in the strong dust condition can be reflected in two aspects. The first is the variability
of the region with the TC track. In June, TC tracks are mostly on the western flanks of
the tropical Atlantic and mostly along the shoreline. In July, less TC activity occurs in
the dust plume region. The second aspect of the suppression is the variability of the
magnitude of TC days and intensity. In August and September, there is an evident
decrease in TC days and intensity in the dust plume region. The monthly variation in
the longitudinal location of the suppression regions presents an eastward migration
from the Gulf of Mexico in June to the eastern tropical Atlantic in September, which
coincides with the response of LWP to the dust semi-direct effect. The monthly
variation of this suppression from June to September presents a weakness, which is
consistent with the weakness in the monthly variation of the westward extension of
the Saharan dust plume.
According to the steady state theory of TCs, a radially and vertically directed
overturning circulation known as the Carnot cycle governs the energy of a TC



(Emanuel, 1986; Rotunno and Emanuel, 1987) and provides an upper bound on the
maximum wind speed in a TC (Emanuel, 1995). In the Carnot cycle, air with
abundant moisture flows into a TC in the boundary layer (Emanuel, 1986). With the
rising motion in a TC, the warm moist air moves upward and cools to saturation as the
temperature decreases. By condensing moisture into cloud and precipitation, the
Carnot cycle converts latent heat to sensible heat to provide the energy of a TC
(Emanuel, 1999). Therefore, the maintenance of a TC apparently depends on the
condensation of moisture into cloud. The Saharan dust plume resides in a thick layer
above the marine boundary layer (Karyampudi and Carlson, 1988). Its semi-direct
effect on LWP can reduce the condensation of moisture into cloud, which suppresses
the maintenance of TC activity.
One notable aspect of the strong dust condition in June is the appearance of TC
activity along 10ºN latitude over the eastern tropical Atlantic, which is consistent with
the response of IWP to the dust microphysical effect (Figure 5). This demonstrates
that Saharan dust can promote TC activity by enhancing IWP through the
microphysical effect. However, this effect is minor in its intensity and area of
influence, compared to the semi-direct effect.
Figure 10 shows the monthly variation in SST, IWP, TC days (on a logarithmic scale),
and TC intensity (on a logarithmic scale) over the region of 20ºW-60ºW, 10ºN-30ºN.
The results indicate the climatology because they are averaged over the 40-yr period
(1980-2019). The monthly variations in both SST and IWP present a linear increase
from June to September. The monthly variations in TC days and intensity (on a
logarithmic scale) present the same linear increase as SST but from July to September,
while the results in June are too small to satisfy this linear relationship. According to
the maximum potential intensity theory (Emanuel, 1987; Holland, 1997), a TC will be
strengthened if SST increases, which has been shown by numerical simulations
applying environmental thermodynamic conditions based on global warming
experiments (Knutson et al., 1998). Therefore, it is the pronounced reduction of SST
caused by strong dust radiative forcing (Figure 7), occurring just over the region



shown in Figure 10, that induces the obvious decease in TC activity in June.

## 6. Summary and conclusions

While the impact of Saharan dust on the atmospheric hydrology and radiation
absorbed by the earth's surface has been documented in previous studies, less
academic attention has been paid to the influence of the Saharan dust plume on
Atlantic TC activity.
In this study, evidence is provided through statistical analyses of various datasets
associated with the Saharan dust plume, atmospheric hydrology, surface temperature,
and Atlantic TC activity over the past 40 years, suggesting that the Saharan dust
plume over the tropical Atlantic can affect TC activity by impacting the atmospheric
hydrology and radiation absorbed by the earth's surface. The influence of the Saharan
dust plume on Atlantic TC activity is complex, and its mechanism and related spatial
and temporal characteristics are summarized as below.
(1) The strong radiative forcing of Saharan dust over the eastern tropical Atlantic in
June is found to produce a pronounced reduction of SST, in the range of 0.2-0.6 °C.
This response of SST to dust radiative forcing helps explain why the duration and
intensity of regional TC activity in June is very small, compared to the increase in TC
activity during the hurricane season (July to September). This mechanism for the
influence of dust on TC activity is climatological in scale because the weakness of
TCs is presented in the 40-yr average from 1980 to 2019, but is evident only in June.
(2) The strengthening Saharan dust plume over the tropical Atlantic during summer
induces the reduction of LWP and IWP through the dust semi-direct effect, which
further suppresses TC activity because the energy of a TC comes from the
condensation of moisture during the Carnot cycle. This suppression of TC activity is
present in the variability of the region with the TC track (mainly in June and July) and
in the variability of the magnitude of TC days and intensity (mainly in August and
September). Both of these show an eastward migration, coinciding with the weakness
in the monthly variation of the westward extension of the Saharan dust plume. This



mechanism for the influence of dust on TC activity occurs when Saharan dust is
intensified and is implemented mainly through LWP because of the stronger dust
semi-direct effect on LWP than on IWP, with the influence covering the whole
Atlantic from the Gulf of Mexico in June to the eastern tropical Atlantic in September.
(3) The strengthening Saharan dust plume is found to enhance IWP along the North
Atlantic ITCZ through the dust microphysical effect, which further promotes TC
activity along 10ºN latitude over the eastern tropical Atlantic in June, by providing
more energy to TCs from moisture condensation. Differentiated from the influence of
dust on TC activity induced by radiative forcing and the semi-direct effect, this
influence of dust on TC activity induced by the microphysical effect is positive, but it
is also minor because it occurs only along the ITCZ over the eastern tropical Atlantic
in June.

**Data availability.**
The MERA-2 monthly dust AOD dataset is available at
https://goldsmr4.gesdisc.eosdis.nasa.gov/data/MERRA2_MONTHLY/M2TMNXAER.
5.12.4/. The MERRA-2 monthly LWP and IWP datasets are available at
https://goldsmr4.gesdisc.eosdis.nasa.gov/data/MERRA2_MONTHLY/M2TMNXCSP.
5.12.4/. The MERRA-2 monthly SST dataset is available at
https://goldsmr4.gesdisc.eosdis.nasa.gov/data/MERRA2_MONTHLY/M2TMNXOC
N.5.12.4/. Atlantic TC track data in 2019 is available at
https://www.nhc.noaa.gov/data/tcr/index.php?season=2019&basin=atl. Atlantic TC
track data (1980-2018) used in this study comes from a global tropical cyclone dataset
which is archived by Massachusetts Institute Technology as a related resource of the
open course "Tropical Meteorology", and located at
ftp://texmex.mit.edu/pub/emanuel/HURR/tracks/. In this dataset, the Atlantic files
were obtained from NOAA's Tropical Prediction Center.



## Author contributions


ZZ carried out the data analysis, led the interpretation of the results, and prepared the
manuscript with contributions from all co-authours. WZ contributed to the
interpretation of the results, provided extensive comments on manuscript, and secured
the funding.

## Competing interests.


The authors declare that they have no conflict of interest.



## Acknowledgments

This work is supported by National Natural Science Foundation of China Grants (41675062, 41375096), and the Research Grants Council of the Hong Kong Special Administrative Region, China (Projects No. CityU 11306417, 11335316).



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



**Table List:**

**Table 1.** The climatology and anomalies of dust AOD averaged over the region of 10º N-25º N, 60º W-20º W. For the extreme conditions, the magnitude of change compared to the climatology is given along with the corresponding percentage.

| Month | 40-yr average (1980-2019) | 8-yr strongest condition | 8-yr weakest condition |
|---|---|---|---|
| 6 | 0.174 | +0.055 (31.70%) | -0.046 (26.44%) |
| 7 | 0.193 | +0.046 (23.78%) | -0.048(24.87%) |
| 8 | 0.149 | +0.031 (20.71%) | -0.041(27.52%) |
| 9 | 0.112 | +0.036 (31.83%) | -0.037(33.04%) |

**Table 2.** The climatology and anomalies of SST (°C) in extreme dust conditions over the region of 10º N-25º N, 60º W-20º W.

| Month | 40-yr average (1980-2019) | 8-yr strongest dust condition | 8-yr weakest dust condition |
|---|---|---|---|
| 6 | 25.48 | -0.31 (1.22%) | +0.01 (0.04%) |
| 7 | 26.10 | -0.07 (0.27%) | -0.01 (0.04%) |
| 8 | 26.83 | -0.10 (0.37%) | -0.09 (0.34%) |
| 9 | 27.31 | -0.13 (0.02%) | +0.09 (0.01%) |



**Figure List:**

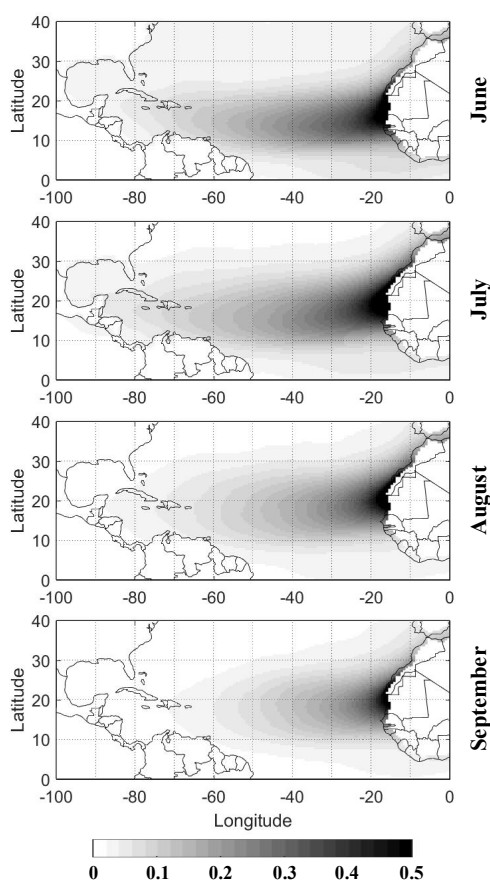

**Figure 1.** 40-yr average (1980-2019) of dust extinction AOD from the MERRA-2 reanalysis product during summer.

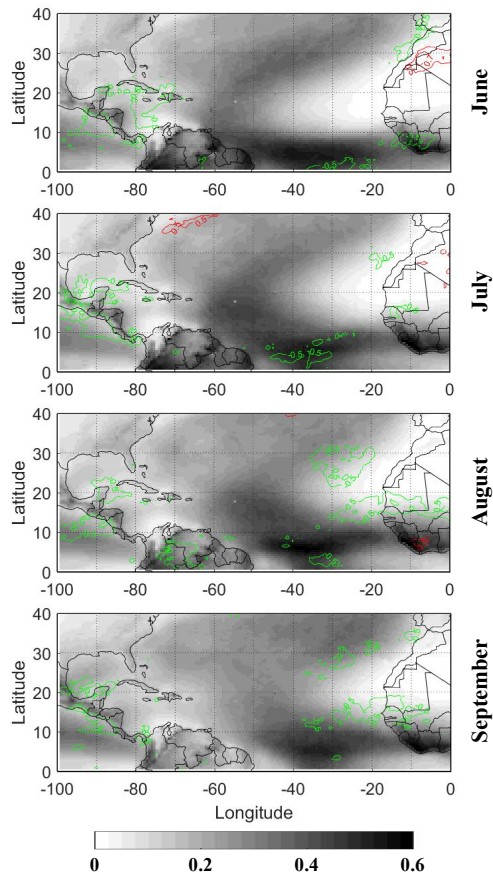

**Figure 2.** 40-yr average (1980-2019) of liquid water path (LWP; kg m$^{-2}$) from the MERRA-2 reanalysis product during summer. The green and red lines are the negative and positive correlation coefficient contours, respectively, for the correlation between LWP and dust AOD at each grid point.

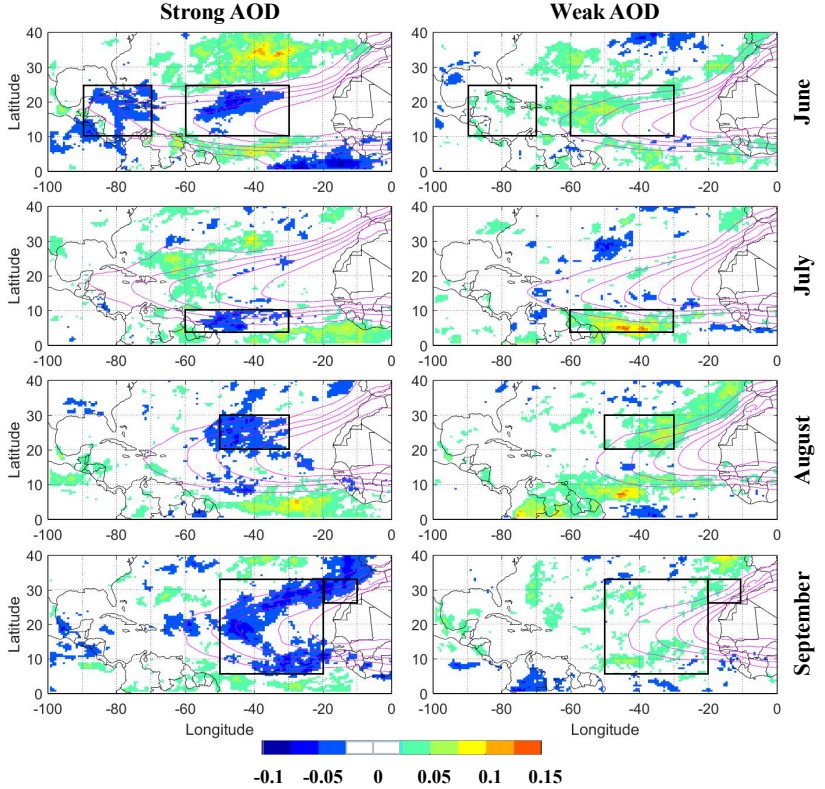

**Figure 3.** (Left column) The difference in LWP (kg m$^{-2}$) between the 8-yr strongest dust AOD
(10ºN-25ºN, 60ºW-20ºW) and the 40-yr average, and (right column) the difference in LWP (kg m$^{-2}$)
between the 8-yr weakest dust AOD and the 40-yr average. Purple lines are the dust AOD
contours of 0.06, 0.1, 0.14, 0.2, and 0.3, averaged for the corresponding 8 years. The solid black
outline shows the negative response region of LWP to dust.



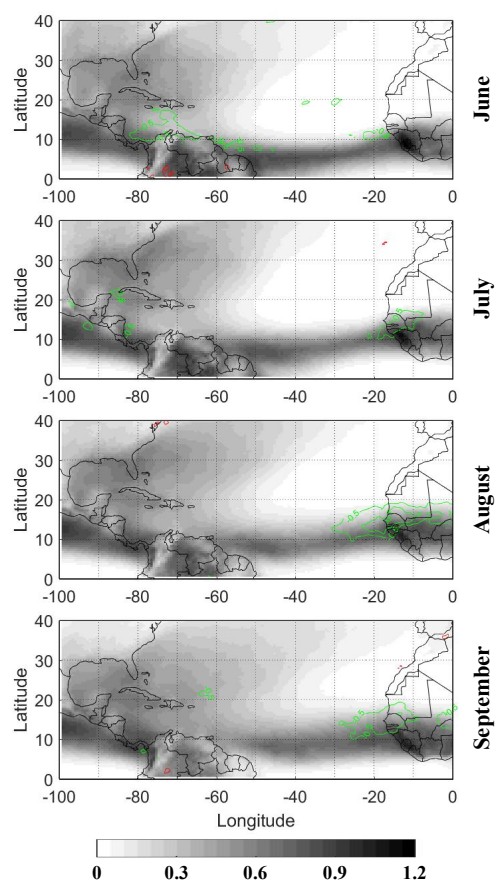

**Figure 4.** 40-yr (1980-2019) average of ice water path (IWP; kg m$^{-2}$) from the MERRA-2 reanalysis product during summer. The green and red lines are the negative and positive correlation coefficient contours, respectively, for the correlation between IWP and dust AOD at each grid point.



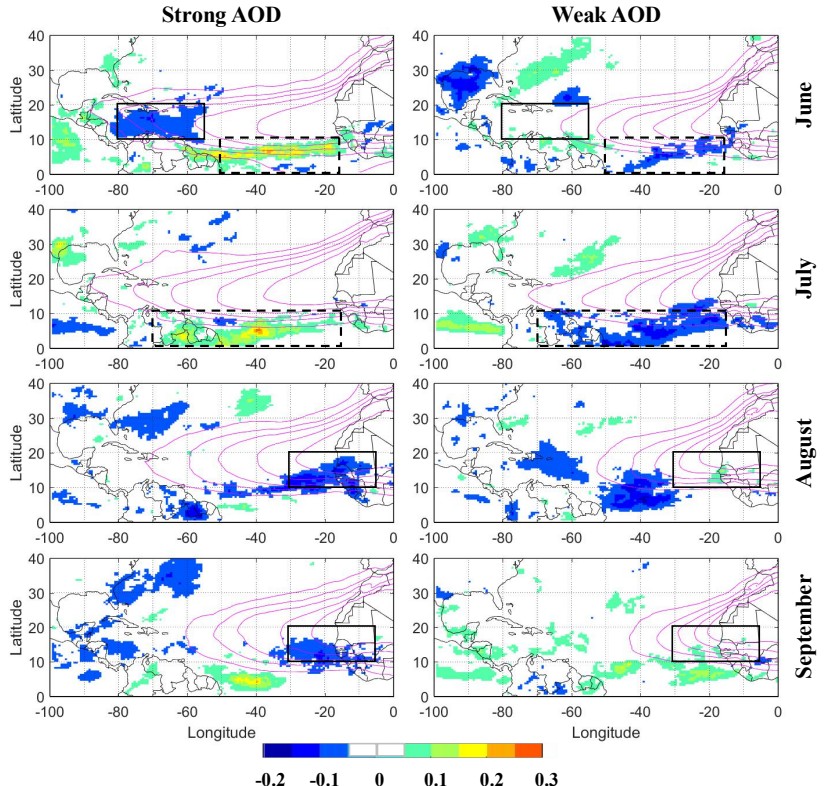

**Figure 5.** (Left column) The difference in IWP (kg m⁻²) between the 8-yr strongest dust AOD
(10ºN-25ºN, 60ºW-20ºW) and the 40-yr average, and (right column) the difference in IWP (kg m⁻²)
between the 8-yr weakest dust AOD and the 40-yr average. Purple lines are the dust AOD
contours of 0.06, 0.1, 0.14, 0.2, and 0.3, averaged for the corresponding 8 years. The solid black
outline shows the negative response region of IWP to dust, and the dashed black outline shows the
positive response region of IWP to dust.



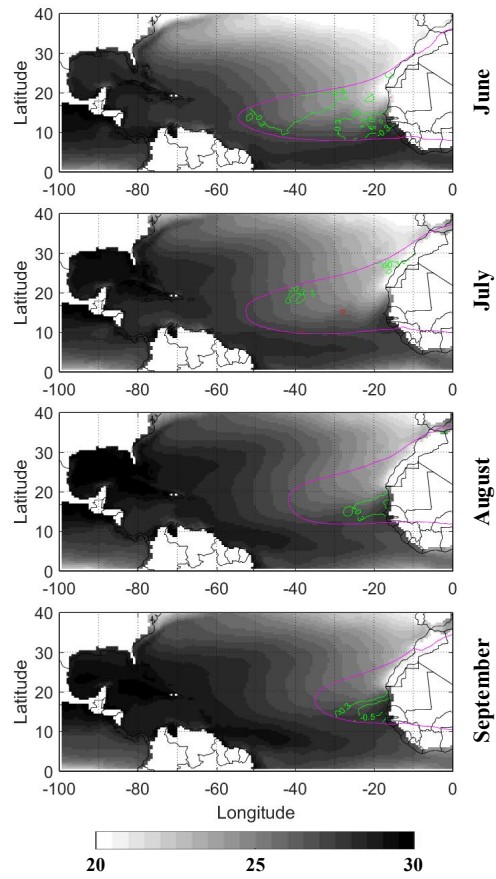

**Figure 6.** 40-yr (1980-2019) average of SST (°C) from the MERRA-2 reanalysis product during summer. The green and red lines are the negative and positive correlation coefficient contours for the correlation between SST and dust AOD at each grid point. Purple lines are the 40-yr (1980-2019) averaged dust AOD contour of 0.15.

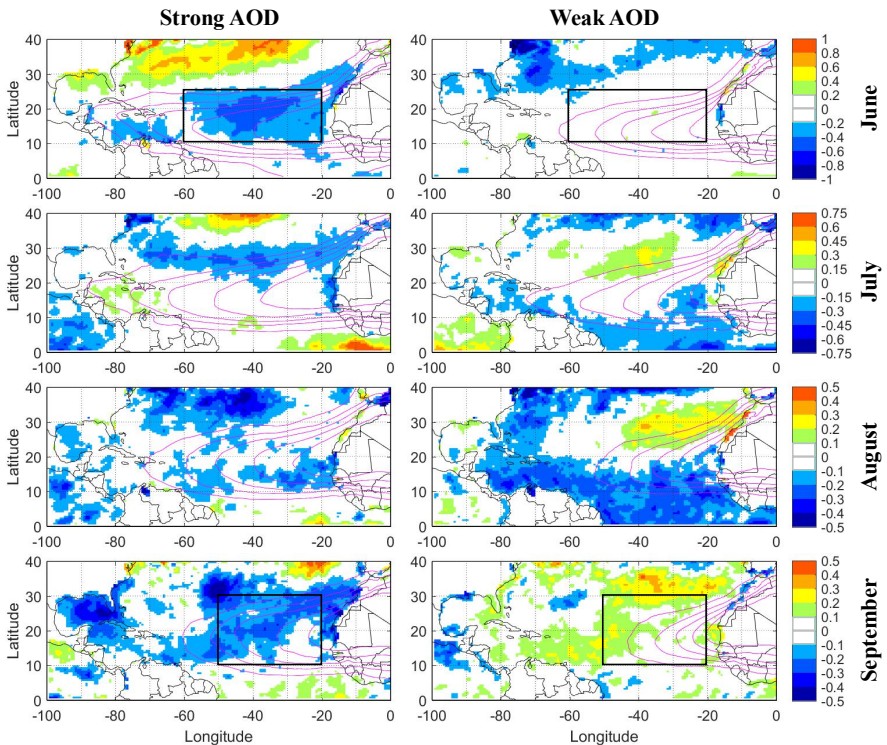

**Figure 7.** (Left column) The difference in SST (°C) between the 8-yr strongest dust AOD
(10ºN-25ºN, 60ºW-20ºW) and the 40-yr average, and (right column) the difference in SST (°C)
between the 8-yr weakest dust AOD and the 40-yr average. Purple lines are the dust AOD
contours of 0.06, 0.1, 0.14, 0.2, and 0.3, averaged for the corresponding 8 years. The solid black
outline shows the negative response region of SST to dust.

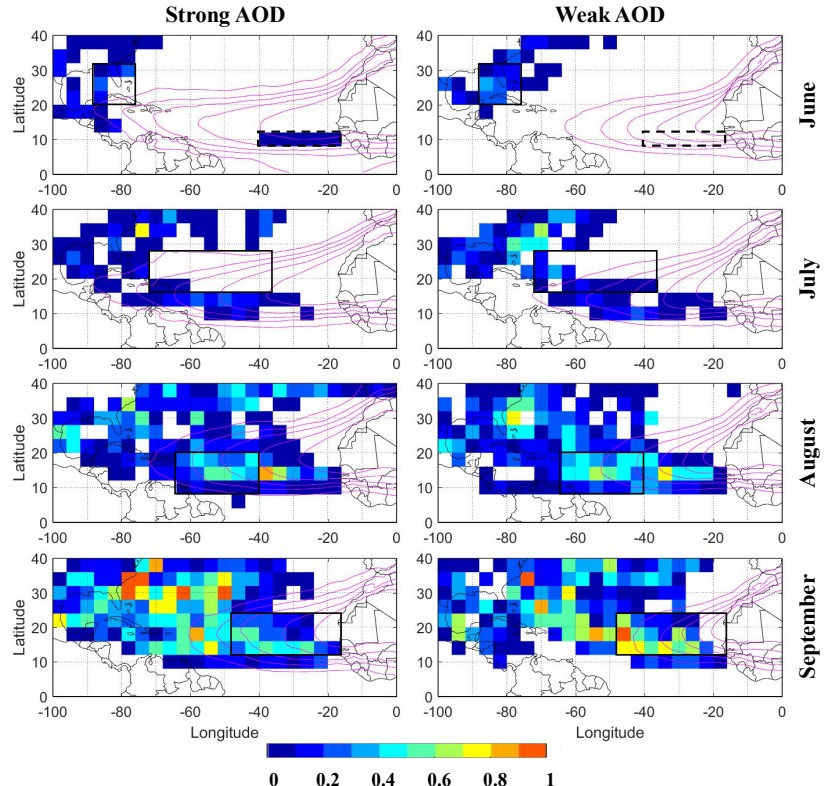

**Figure 8.** Comparison of TC days at 4-degree grid resolution between the 8-yr strongest and

weakest dust conditions. The images represent annual average values. Purple lines are the dust

AOD contours of 0.06, 0.1, 0.14, 0.2, and 0.3, averaged for the corresponding 8 years. The solid

black outline shows the negative response region of TC days to dust, and the dashed black outline

shows the positive response region of TC days to dust.



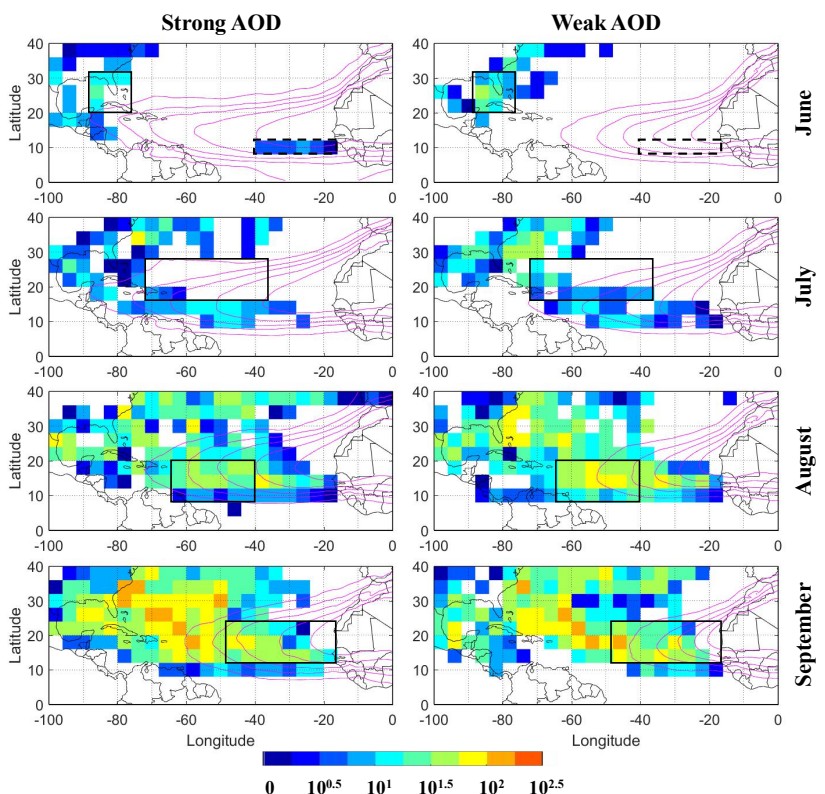

**Figure 9.** Same as Figure 8, but for TC intensity (m s$^{-1}$).



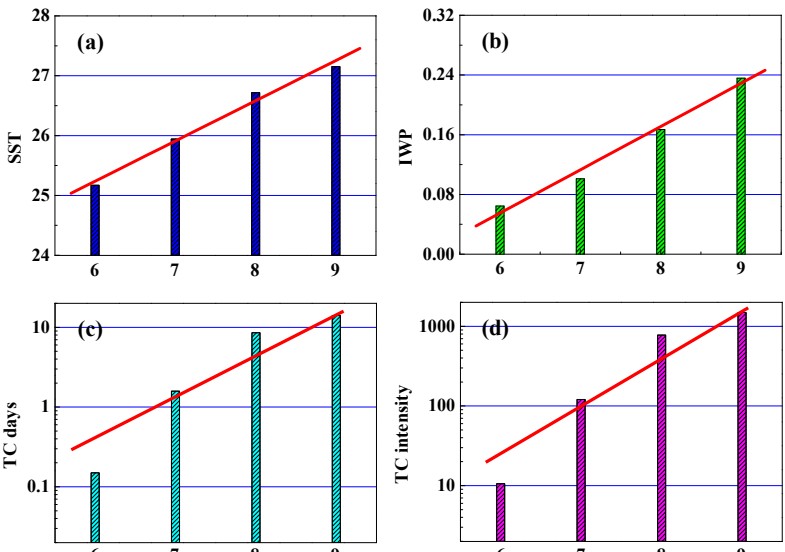

**Figure 10.** Monthly variation (from June to September) of (a) SST (°C), (b) IWP (kg m⁻²), (c) TC
days, and (d) TC intensity (m s⁻¹) over the region of 20ºW-60ºW, 10ºN-30ºN, averaged during the
40-yr period (1980-2019). The red boldface lines in the SST and IWP panels indicate the
least-squares best fit line to the data, and the linear increase tendency in the data. The red boldface
lines in the TC days and intensity panels have the same linear increase tendency as SST.
