# Peer review of "Influence of Saharan dust on Atlantic tropical cyclones"

_Atmospheric Chemistry and Physics, 2020_

## Referee Comment (RC1) · Anonymous Referee #1 · 9 Sep 2020

The manuscript attempts to investigate the effect of Saharan dust on tropical cyclones. The topic is definitely interesting, but I am deeply concerned with the methodology adopted in this study which is mainly based on the MERRA2 reanalysis. The result analyses seem superficial as well, which fails to reveal the physical linkage between dust and TC. I will further elaborate those two concerns below. In its present form, I cannot recommend publication of this work by ACP.

1) Inappropriate use of reanalysis data. It's well known that reanalysis data like MERRA2 assimilate available observations as much as possible to achieve the best performance. As a result, the different variables in MERRA2, including AOD, SST, LWP, and IWP used in this study, may not be linked physically. The different variables also have different sources of observations with different time/spatial coverages to be

assimilated, which further impairs the physical relationship among them. Take AOD for example, what are the observational constraints it has before the Terra/Aqua Era (before 1999)? The manuscript's analyses largely rely on the 8-yr strongest and weakest AOD contrast during 1980-2019, but the AOD record during those forty years are of great uncertainty. LWP and IWP in MERRA2 are considered to have even larger uncertainty than AOD, largely owing to the crude convection parameterizations.

2) Too simply analyses. The manuscript assesses the dust impacts via the comparisons of different time periods/conditions with different dust loadings. However, this way cannot rule out the other covarying factors and climate natural variability. Also, the current analyses cannot imply causation for those two-way interactions. For example, AOD can affect SST by interfering atmospheric radiation, but SST can also influence dust outflow by regulating the large-scale circulations. If the authors want to verify and explain the relationships they obtained between AOD and SST/TC/cloud properties, I strongly suggest them perform free-runs of a GCM, ideally the same with MERRA2 uses.

---

## Referee Comment (RC2) · Anonymous Referee #2 · 14 Sep 2020

General comments: Sorry to say that this manuscript is not suitable for publication in the present form on a scientific journal for the following reasons: -The shown correlations do not imply causation. The authors didn't put any effort in trying to demonstrate the mechanism behind their assumptions: the sentence "A causes something to B BECAUSE they are highly negatively correlated each other" (see line 38 and many other places in the manuscript) doesn't make any sense and it is at the base of the current manuscript. - The time period used is not clear: why only back to 1980? - correlation are shown without considering statistical significance - diffferences are shown without any consideration regarding the statistical significance of the shown results: you can't just put to "color white" values lower than a threshold to highlight interesting patterns. - You can find thousand of series showing the same behaviour of figure 5.. once again

causation is missed.

- There are many minor comments such as : suspended phrases, missed units, inconsistent colorbars, etc.

---

## Author Comment (AC1) · 20 Sep 2020

Q1: As a result, the different variables in MERRA2, including AOD, SST, LWP, and IWP used in this study, may not be linked physically.

Answer: Briefly, MERRA-2 is produced using the GEOS-5 atmospheric general circulation model (Gelaro et al., 2017) and data assimilation system version 5.12.4 (Rienecker et al. 2008; Molod et al. 2015) and the Three-dimensional Variational Data Analysis (3DVAR) Gridpoint Statistical Interpolation (GSI) meteorological analysis scheme (Wu et al. 2002; Kleist et al. 2009).

The variables in data assimilation system may not be linked physically while they are linked physically in the GEOS-5 atmospheric general circulation model. These physi-

cal linkages are presented through the physics package of the GEOS-5 atmospheric general circulation model, including four major groups of physical processes: moist processes, radiation, turbulent mixing, and surface processes (Rienecker et al. 2008). Each of these in turn is subdivided into various components. The radiation module includes longwave and shortwave radiation submodules. The turbulent mixing consists of the vertical diffusion, planetary boundary layer parameterization, and gravity wave drag. The surface processes provide surface fluxes obtained from land, ocean and sea ice models. To minimize the spurious periodic perturbations of the analysis, MERRA uses the Incremental Analysis Update (IAU) technique developed by Bloom et al. (1996). In MERRA-2 the concatenation of the IAU corrector segments of each analysis cycle is equivalent to a single, continuous run of the atmospheric general circulation model (Bosilovich et al., 2016). Budgets in MERRA-2 are thus identical to those in the free-running atmospheric general circulation model (Bosilovich et al., 2016). The MERRA-2 output includes a nearly full accounting for the budgets of atmospheric quantities: the mass of the atmosphere, the mass of water in vapor, liquid, and ice forms, kinetic energy, the virtual enthalpy, virtual potential temperature, aerosol, and the total mass of odd oxygen (Bosilovich et al., 2016).

The budgets of atmospheric quantities represent the physical relationship among the MERRA-2 output. The MERRA-2 output used in this study is related mainly to the budget of atmospheric water, total energy of the atmospheric column, and potential temperature.

References:

Bloom, S., L. Takacs, A. DaSilva, and D. Ledvina, 1996: Data assimilation using incremental analysis updates. Mon. Wea. Rev., 124, 1256-1271.

Bosilovich, M. G., R. Lucchesi, and M. Suarez, 2016: MERRA-2: File Specification. GMAO Office Note No. 9 (Version 1.1), 73 pp, available from http://gmao.gsfc.nasa.gov/pubs/office_notes.

Gelaro, R., and Coauthors, 2017: The Modern-Era Retrospective Analysis for Research and Applications, Version 2 (MERRA-2). Journal of Climate, 30(14), doi: 10.1175/JCLI-D-16-0758.1.

Kleist, D. T., D. F. Parrish, J. C. Derber, R. Treadon, W.-S. Wu, , and S. Lord, 2009: Introduction of the GSI into the NCEP Global Data Assimilation System. Weather Forecasting, 24, 1691–1705.

Molod, A. M., L. L. Takas, M. Suarez, and J. Bacmeister, 2015: Development of the GEOS-5 atmospheric general circulation model: Evolution from MERRA to MERRA-2. Geoscientific Model Development, 8, 1339–1356, doi:10.5194/gmd-8-1339-2015.

Rienecker, M. M., and Coauthors, 2008: The GEOS-5 Data Assimilation System—-Documentation of versions 5.0.1, 5.1.0, and 5.2.0. Technical Report Series on Global Modeling and Data Assimilation, Vol. 27, NASA Tech. Rep. NASA/TM-2008-104606, 118 pp. [Available online at https://gmao.gsfc.nasa.gov/pubs/docs/Rienecker369.pdf.]

Wu, W.-S., R. Purser, and D. Parrish, 2002: Three-dimensional variational analysis with spatially inhomogeneous covariances. Mon. Wea. Rev., 130, 2905–2916.

————————————————————————————————————————————-

Q2: The different variables also have different sources of observations with different time/spatial coverages to be assimilated, which further impairs the physical relationship among them.

Answer: The research in our study is related mainly to the moist physics. Gelaro et al. (2017) introduce the analysis algorithm of moisture field in MERRA-2. The control variable for moisture used in MERRA-2 is the normalized pseudorelative humidity (Holm 2003) defined by the pseudorelative humidity scaled by its background error standard deviation. The normalized pseudorelative humidity has a near Gaussian error distribution, making it more suitable for the minimization procedure employed in the assimilation scheme. Also within the GSI, a tangent linear normal mode constraint

(Kleist et al. 2009) is applied during the minimization procedure to control noise and improve the overall use of observations. The background error statistics used in the GSI have been updated as well in MERRA-2, which exhibit generally smaller standard deviations for most variables compared with the MERRA system, but both larger and smaller correlation length scales depending on the variable, latitude, and vertical level.

Besides moisture field, other MERRA-2 fields, including the meteorological, radiation, ozone, and cryospheric fields, have been validated, that are detailed in Bosilovich et al. (2016).

References:

Bosilovich, M. G., and Coauthors, 2016: MERRA-2: Initial evaluation of the climate. Technical Report Series on Global Modeling and Data Assimilation NASA/TM-2015-104606 43, NASA Global Modeling and Assimilation Office. URL https://gmao.gsfc.nasa.gov/pubs/tm/docs/Bosilovich803.pdf.

Gelaro, R., and Coauthors, 2017: The Modern-Era Retrospective Analysis for Research and Applications, Version 2 (MERRA-2). Journal of Climate, 30(14), doi: 10.1175/JCLI-D-16-0758.1.

Holm, E. V., 2003: Revision of the ECMWF humidity analysis: Construction of a Gaussian control variable. Proc. ECMWF/GEWEX Workshop on Humidity Analysis, Reading, United Kingdom, ECMWF/GEWEX, 6 pp. [Available online at http://www.ecmwf.int/sites/default/files/elibrary/2003/9998-revision-ecmwf-humidity-analysis-construction-gaussian-controlvariable.pdf.]

Kleist, D. T., D. F. Parrish, J. C. Derber, R. Treadon, R. M. Errico, and R. Yang, 2009a: Improving incremental balance in the GSI 3DVAR analysis system. Mon. Wea. Rev., 137, 1046–1060, doi:10.1175/2008MWR2623.1.

——————————————————————————————————————————

Q3: Take AOD for example, what are the observational constraints it has before the

Terra/Aqua Era (before 1999)?

Answer: The MERRA-2 aerosol analysis uses the Goddard Aerosol Assimilation System (GAAS) (Buchard et al., 2015, 2016), which assimilates quality-controlled AOD at 550 nm into the GEOS-5/GOCART modeling system. The AOD observations used in MERRA-2 includes AVHRR (1980 – August 2002), AERONET (Station dependent, 1999 – October 2014), MISR (February 2000 – June 2014), and MODIS (Terra: March 2000 – onwards; Aqua: August 2002 – onwards) (Randles et al., 2017).

References:

Buchard, V., and Coauthors, 2015: Using the OMI aerosol index and absorption aerosol optical depth to evaluate the NASA MERRA Aerosol Reanalysis. Atmospheric Chemistry and Physics, 15 (10), 5743–5760, doi:10.5194/acp-15-5743-2015.

Buchard, V., and Coauthors, 2016: Evaluation of the surface PM2.5 in Version 1 of the NASA MERRA Aerosol Reanalysis over the United States. Atmospheric Environment, 125, 100–111, doi:10.1016/j.atmosenv.2015.11.004.

Randles, C. A., A. M. da Silva, V. Buchard, P. R. Colarco, A. Darmenov, R. Govindaraju, A. Smirnov, B. Holben, R. Ferrare, J. Hair, Y. Shinozuka, and C. J. Flynn, 2017: The MERRA-2 aerosol reanalysis, 1980–onward, Part I: System description and data assimilation evaluation, Journal of Climate, 30(17), 6823–6850.

———————————————————————————————————————————-

Q4: The manuscript's analyses largely rely on the 8-yr strongest and weakest AOD contrast during 1980-2019, but the AOD record during those forty years are of great uncertainty.

Answer: Because the AOD in MERRA-2 has different sources of observations with different time and spatial coverages to be assimilated, the progress of bias correlation is implemented in MERRA-2. The bias-corrected approach involves cloud screening and homogenization of the observing system by means of a neural net retrieval (NNR) that

translates cloud-cleared observed radiances into AERONET-calibrated AOD (Randles et al., 2017).

To reduce errors, analysis of AOD ($\tau$a) is performed using error covariances derived from innovation data using the maximum-likelihood method of Dee and da Silva (1999). The AOD analysis equation (Randles et al., 2017) is:

$$\tau a = \tau f + [H*Pf*HT/(H*Pf*HT+R)]*(\tau o - H*xf) \quad (1)$$

where the superscripts a, f, and o indicate the analysis, forecast (background), and observation, respectively. H is the linear observation operator that converts aerosol mass to AOD. The operators Pf and R are the background and observation error covariance matrices, respectively. x is aerosol mass mixing ratio. Forecast of AOD ($\tau$f) and aerosol mass mixing ratio (xf) is performed using the MERRA-2 modeling system relevant for the aerosol (GEOS-5 coupled to the GOCART aerosol module).

References:

Dee, D. P., and A. M. da Silva, 1999: Maximum-likelihood estimation of forecast and observation error covariance parameters. Part I: Methodology. Monthly Weather Review, 127, 1811–1834.

Randles, C. A., A. M. da Silva, V. Buchard, P. R. Colarco, A. Darmenov, R. Govindaraju, A. Smirnov, B. Holben, R. Ferrare, J. Hair, Y. Shinozuka, and C. J. Flynn, 2017: The MERRA-2 aerosol reanalysis, 1980–onward, Part I: System description and data assimilation evaluation, Journal of Climate, 30(17), 6823–6850.

———————————————————————————————————————————————–

Q5: LWP and IWP in MERRA2 are considered to have even larger uncertainty than AOD, largely owing to the crude convection parameterizations.

Answer: Since MERRA, the GEOS model has undergone changes to both its dynamical core and its physical parameterizations. For the convection parameterization, the

MERRA-2 model includes a Tokioka-type trigger on deep convection as part of the Relaxed Arakawa–Schubert convective parameterization scheme (Moorthi and Suárez 1992), which governs the lower limit on the allowable entrainment plumes (Bacmeister and Stephens 2011). Moreover, upgrades to the moist physical parameterization schemes in MERRA-2 model also include increased reevaporation of frozen precipitation and cloud condensate (Molod et al. 2015), which is helpful to reduce the uncertainty in the calculation of LWP and IWP. Cloud parameterization scheme in MERRA-2 model includes the source terms for cloud, anvil cloud, large-scale condensation, freezing and melting of cloud condensate, evaporation cloud, autoconversion of liquid and mixed phase cloud, sedimentation of ice cloud, and fallout and re-evaporation of precipitation and accretion of cloud condensate (Rienecker et al., 2008).

References:

Bacmeister, J. T., and G. Stephens, 2011: Spatial statistics of likely convective clouds in CloudSat data. J. Geophys. Res., 116, D04104, doi:10.1029/2010JD014444.

Molod, A., L. Takacs, M. Suárez, and J. Bacmeister, 2015: Development of the GEOS-5 atmospheric general circulation model: Evolution from MERRA to MERRA2. Geosci. Model Dev., 8, 1339–1356, doi:10.5194/gmd-8-1339-2015.

Moorthi, S., and M. J. Suárez, 1992: Relaxed Arakawa–Schubert: A parameterization of moist convection for general circulation models. Mon. Wea. Rev., 120, 978–1002, doi:10.1175/1520-0493(1992)120,0978:RASAPO.2.0.CO;2.

Rienecker, M. M., and Coauthors, 2008: The GEOS-5 Data Assimilation System—-Documentation of versions 5.0.1, 5.1.0, and 5.2.0. Technical Report Series on Global Modeling and Data Assimilation, Vol. 27, NASA Tech. Rep. NASA/TM-2008-104606, 118 pp. [Available online at https://gmao.gsfc.nasa.gov/pubs/docs/Rienecker369.pdf.]

—————————————————————————————————————————-

Q6: If the authors want to verify and explain the relationships they obtained between

[Figure]

AOD and SST/TC/cloud properties, I strongly suggest them perform free-runs of a GCM, ideally the same with MERRA2 uses.

Answer: Please refer to the answer of question 1.

Here it is noted that MERRA-2 is produced with the GEOS atmospheric data assimilation system. The key components of the system includes the GEOS-5 atmospheric model (Rienecker et al. 2008; Molod et al. 2015), an atmospheric general circulation model. In MERRA-2 the concatenation of the IAU corrector segments of each analysis cycle is equivalent to a single, continuous run of the AGCM (Bosilovich et al., 2016). Budgets in MERRA-2 are thus identical to those in the free-running AGCM. The MERRA-2 output includes a nearly full accounting for the budgets of atmospheric quantities, including the mass of water (in vapor, liquid, and ice forms) and virtual potential temperature. Following the suggestion of referee, we are going to perform free-runs of a GCM, and its results will be considered as the supplement of this manuscript.

References:

Bosilovich, M. G., R. Lucchesi, and M. Suarez, 2016: MERRA-2: File Specification. GMAO Office Note No. 9 (Version 1.1), 73 pp, available from http://gmao.gsfc.nasa.gov/pubs/office_notes.

Molod, A. M., L. L. Takas, M. Suarez, and J. Bacmeister, 2015: Development of the GEOS-5 atmospheric general circulation model: Evolution from MERRA to MERRA-2. Geoscientific Model Development, 8, 1339–1356, doi:10.5194/gmd-8-1339-2015.

Rienecker, M. M., and Coauthors, 2008: The GEOS-5 Data Assimilation System—-Documentation of versions 5.0.1, 5.1.0, and 5.2.0. Technical Report Series on Global Modeling and Data Assimilation, Vol. 27, NASA Tech. Rep. NASA/TM-2008-104606, 118 pp. [Available online at https://gmao.gsfc.nasa.gov/pubs/docs/Rienecker369.pdf.]

Please also note the supplement to this comment:

[Figure]

https://acp.copernicus.org/preprints/acp-2020-761/acp-2020-761-AC1-supplement.pdf

---

## Author Comment (AC2) · 30 Sep 2020

Q1-The shown correlations do not imply causation. The authors didn't put any effort in trying to demonstrate the mechanism behind their assumptions: the sentence "A causes something to B BECAUSE they are highly negatively correlated each other" (see line 38 and many other places in the manuscript) doesn't make any sense and it is at the base of the current manuscript.

Answer: Although this study presents the relationships obtained between dust AOD and SST/TC/cloud properties, the physical mechanism behind these correlations has not been studied in depth, which will be the focus of our next research step, and the results will be used as a supplement to this manuscript.

[Figure]

———————————————————————————————————————-

Q2- The time period used is not clear: why only back to 1980?

Answer: MERRA-2 data starts from 1980.

———————————————————————————————————————-

Q3- correlation are shown without considering statistical significance

Answer: The statistical significance of correlation is 95%, and we will change the relevant text content.

———————————————————————————————————————-

Q4- diffferences are shown without any consideration regarding the statistical significance of the shown results: you can't just put to "color white" values lower than a threshold to highlight interesting patterns.

Answer: According to the comment of referee, we will reproduce the pictures, and change the relevant text content.

———————————————————————————————————————-

Q5- You can find thousand of series showing the same behaviour of figure 5.. once again causation is missed.

Answer: Please refer to the answer of Q1.

———————————————————————————————————————-

Q6- There are many minor comments such as : suspended phrases, missed units, inconsistent colorbars, etc.

Answer: These questions will be modified in manuscript.